# H-Bonding Room Temperature Phosphorescence Materials via Facile Preparation for Water-Stimulated Photoluminescent Ink

**DOI:** 10.3390/molecules27196482

**Published:** 2022-10-01

**Authors:** Lingyun Lou, Tianqi Xu, Yuzhan Li, Changli Zhang, Bochun Wang, Xusheng Zhang, Hean Zhang, Yuting Qiu, Junyan Yang, Dong Wang, Hui Cao, Wanli He, Zhou Yang

**Affiliations:** Department of Materials Physics and Chemistry, School of Materials Science and Engineering, University of Science and Technology Beijing, Xueyuan Road 30#, Haidian District, Beijing 100083, China

**Keywords:** room temperature phosphorescence, H-bond, water-stimulated

## Abstract

Pure organic room-temperature phosphorescence (RTP) materials built upon noncovalent interactions have attracted much attention because of their high efficiency, long lifetime, and stimulus-responsive behavior. However, there are limited reports of noncovalent RTP materials because of the lack of specific design principles and clear mechanisms. Here, we report on a noncovalent material prepared via facile grinding that can emit fluorescence and RTP emission differing from their components’ photoluminescent behavior. Exciplex can be formed during the preparation process to act as the minimum emission unit. We found that H-bonds in the RTP system provide restriction to nonradiative transition but also enhance energy transformation and energy level degeneracy in the system. Moreover, water-stimulated photoluminescent ink is produced from the materials to achieve double-encryption application with good resolution.

## 1. Introduction

Pure organic room temperature phosphorescence (RTP) has become an attractive way to generate long afterglow photoluminescence in the past decade. The phosphorescence emission of organic molecules is significantly influenced by intersystem crossing (ISC) from singlet state to triplet state and is impacted by nonradiative transition. Since heavy metal atoms can effectively promote the ISC process, most organic RTP materials developed are based on organometallic coordination complex. However, organometallic compounds with heavy metal coordination center atoms are costly to produce and environmentally hazardous [1,2,3]. Pure organic RTP materials are excellent substitutes without the defects mentioned above [4,5]. Many efforts have been made to facilitate the transition from organometallic to pure organic RTP materials, including synthesizing molecules capable of generating long lifetime triplet excitons [6,7,8], designing new polymer-based and host-guest organic RTP systems [9,10,11,12,13,14], and completing the mechanism of pure organic RTP emission [15,16]. Consequently, pure organic RTP materials now can achieve ultra-long-lifetime, high phosphorescent quantum yield, good biocompatibility, and stimulus-responsive luminescence and even make breakthroughs in electroluminescent RTP emission [17,18,19,20,21,22].

The most common pure organic RTP systems are polymer matrix RTP materials and crystalline RTP materials [23,24,25]. In these materials, the phosphorescent molecules are restricted within a rigid structure, which can mitigate the nonradiative relaxation and annihilation of triplet excitons. However, the rigid structure also restricts the ability of phosphors to respond to an external stimulus. Theoretically, the RTP emission of pure organic molecules can be easily affected by ambient conditions such as oxygen, water, and temperature. Unfortunately, it is hard for polymer-based and crystalline RTP materials to utilize this stimuli-responsive behaviour. To make breakthroughs in the development of stimuli-responsive RTP materials, dynamic and spontaneous noncovalent interactions are often introduced. Among noncovalent interactions, the H-bond draws increasing attention in RTP systems because it endows pure organic RTP materials with the properties of self-assembly, energy transference, and ambient responsiveness. RTP emission of these materials can be adjusted by tailoring the H-bonds [26,27,28,29].

Carbazole and its derivatives are promising RTP materials in crystalline state and in polymer matrices. Most of its photoluminescent derivatives possess functional groups attached to the N atom [30,31]. Our group has reported two carbazole derivatives with functional groups linked to the benzene structure in carbazole and has successfully achieved efficient RTP emission by doping them in polymer matrices [32,33]. Based on our previous work, we designed H-bonded double-component RTP materials capable of forming exciplex to obtain similar fluorescence and phosphorescence emissions. A carbazole derivative 4,4’-(9H-carbazole-2,7-diyl)dibenzoic acid (DBAc-Cz) with carboxyl groups was synthesized. To construct H-bond based organic RTP materials, a heterocyclic compound 1,5,7-Triazabicyclo [4.4.0]dec-5-ene (TBD) was used [34,35]. The R_2_NH and R_3_N groups not only serve as H-bond supporter but can transfer energy intermolecularly and significantly affect the photoluminescent properties, allowing for room-temperature phosphorescence. Based on this design, pure organic noncovalent materials with an emission RTP up to 0.85 s lifetime and with a maximum phosphorescence quantum yield of 11.71% were successfully developed. The noncovalent connecting process happened spontaneously when DBAc-Cz and TBD were ground, which produced exciplex with fluorescence and RTP behavior different from the two components. Water-stimulated ink was prepared by utilizing the properties of H-bonds and was tested on a paper to demonstrate color-changeable fluorescence and water-erasable RTP emission.

## 2. Results and Discussion

The chemical structure of DBAc-Cz and TBD is presented in Figure 1a and the synthesis route of DBAc-Cz is shown in Appendix A. The conjugate structure of carbazole offers DBAc-Cz photoluminescent features, and the symmetrical carboxyl groups are designed for H-bond interaction (Figure 1a). DBAc-Cz is a photoluminescent molecule under UV radiation, while TBD almost has no photoluminescent emission (Figure 1b). In addition, both molecules have no phosphorescence at room temperature. To construct noncovalent RTP materials, DBAc-Cz and TBD were processed by the facile grinding method. Two molecules in powder state were thoroughly ground, and the resulting powder showed fluorescence and RTP emission varying significantly from the two ingredients (Figure 1b). Four mixtures with different molar ratio of DBAc-Cz and TBD were prepared and characterized, including DBAc-Cz:TBD = 1:500 (noted as DT500), DBAc-Cz:TBD = 1:100 (noted as DT100), DBAc-Cz:TBD = 1:50 (noted as DT50), and DBAc-Cz:TBD = 1:20 (noted as DT20). These materials exhibited bright fluorescent and phosphorescence emission with up to 6 s macroscopic RTP afterglow (Figure 1c).

Encouraged by these results, detailed photoluminescence (PL) characterization was carried out to systematically study the materials. DBAc-Cz showed a blue fluorescence emission peak around 461 nm, while TBD showed almost no emission at all according to the black curve (Figure 2a). At room temperature, DBAc-Cz and TBD had no phosphorescence emission. DBAc-Cz showed phosphorescence emission at 77 K with two peaks around 528 nm and 572 nm (Figure 2a). Interestingly, when DBAc-Cz and TBD were ground thoroughly to produce a powder mixture noted as DT, photoluminescence emission changed greatly. The Commission Internationale de L’Eclairage (CIE) chromaticity coordinate of DBAc-Cz’s fluorescence emission was calculated to be (0.2296, 0.2755). As shown in Figure 2b, the fluorescence emission peaks of DT500, DT100, DT50, and DT20 were observed at 507 nm, 511 nm, 519 nm, and 525 nm, respectively. The CIE chromaticity coordinates for the DTs were calculated to be (0.2571, 0.4287), (0.2789, 0.4326), (0.2965, 0.4523), and (0.3089, 0.4565), respectively. The DTs’ fluorescence emissions were close to each other and distinctly differed from that of DBAc-Cz (Figure 2b and Appendix A). In addition, DTs showed RTP emissions, with maximum emission wavelengths around 529 nm (DT500), 537 nm (DT100), 533 nm (DT50), and 539 nm (DT20). The differences in the maximum emission wavelengths between fluorescence and RTP were 22 nm (DT500), 26 nm (DT100), 14 nm (DT50), and 16 nm (DT500) (Table 1 and Appendix A). Neither TBD nor DBAc-Cz showed RTP emission. TBD is not a photoluminescent molecule, and DBAc-Cz’s 77 K characteristic phosphorescence emission with double peaks and the waveform is different from DT’s RTP emission (Figure 2c). The variation in the photoluminescent features also exists on the molecule our group synthesised of 4,4’-(9H-carbazole-2,7-diyl)diphenol (noted as DPOl-Cz) when ground with TBD (Appendix A). DPOl-Cz‘s chemical structure is close to DBAc-Cz only with the hydroxy groups replaced by carboxyl groups. DPOl-Cz showed almost no photoluminescent emission under 365 nm UV radiation. When DPOl-Cz was ground together with TBD in the same way to produce DTs, the mixture exhibited emission around 483 nm (Appendix A).

Based on the photoluminescence characterization results, it was concluded that the changed fluorescence emission and the RTP emission of DTs were not due to DBAc-Cz or TBD but from another unit inside the DTs. Therefore, UV-vis spectroscopy was used to further investigate the DTs. All the samples were dissolved in dichloromethane (DCM) for the test. TBD’s characteristic absorption peak was around 292 nm, which vanished in DT100′s, DT50′s, and DT20′s absorption spectra. The spectra of these three DTs showed absorption waveforms different from DBAc-Cz except the absorption peak around 236 nm (Figure 2d). The absorption spectrum of DT500 showed a peak around 292 nm, which was also observed for TBD. This indicated that microscale DBAc-Cz in the DTs resulted in significant changes in the UV-vis absorption features, reflecting the energy changes between DBAc-Cz and TBD within the DTs. The differences in UV-vis absorption spectra and PL/phosphorescence spectra (Figure 2b–d) between DTs and two starting components suggested that a newly formed unit existed in DTs was responsible for the RTP behavior.

Double emission (fluorescence and RTP) photoluminescent materials with close fluorescence and RTP maximum emission wavelength could be obtained after TBD and DBAc-Cz were simply ground together. In addition to this feature, DTs showed excellent RTP emission lifetime with a phosphorescence quantum yield of up to 11.71% (DT500). DT500, DT100, DT50, and DT20 exhibited RTP with lifetime of 0.79 s, 0.84 s, 0.81 s, and 0.85 s, respectively. (Figure 3a–d). Although the fitting curves for determining the lifetime of DTs were similar, the visual afterglows of DTs differed from each other. It was found that the mole ratio caused differences in RTP afterglow and influenced the emission intensity. DT50 showed the strongest photoluminescence or fluorescence intensity, while DT500 exhibited the finest RTP emission intensity and afterglow among four materials (Figure 1c and Figure 3e,f).

While the grinding method provided an easy way to prepare these RTP materials, the DTs were not homogeneous, so specific structures of the newly formed units inside them were difficult to analyze. This can be observed in the photographs obtained with the fluorescence microscope (Figure 4g). The morphology of the DT powder presented TBD-like small crystals. In DT, amorphous DBAc-Cz powder was considered to be surrounded by TBD. The majority of crystal-like DT powder showed bright emission regardless the changed ratio of DBAc-Cz and TBD. Still, a small quantity of non-photoluminescent TBD not contacted with DBAc-Cz could be observed in DT (marked by a white ellipse). It was hard to define how the DBAc-Cz were surrounded and connected with TBD to form units inside DT with the distinct fluorescent and phosphorescent properties. Therefore, characterizations of noncovalent interaction and energy level were performed to explain these units. Firstly, FT-IR spectra were measured to test the intermolecular H-bond between DBAc-Cz and TBD [36,37]. Theoretically, the two carboxyl groups of DBAc-Cz and the R_2_NH and R_3_N groups of TBD can form intermolecular H-bond. As shown in Figure 4a, the IR spectra of DT showed a wide peak around 2640–2770 cm^−1^ compared with that of DBAc-Cz and TBD. The sharp peaks around 1655 cm^−1^ of DT were attributed the shifted peak of TBD around 1640 cm^−1^. These two phenomena indicated the existence of intermolecular H-bond between DBAc-Cz and TBD, which reduced the distance between the two molecules and was the prerequisite for forming exciplex. To further demonstrate the formation of exciplex in DT, UV-vis diffuse reflection experiment was performed. The UV-vis diffuse reflection spectra of DT were very similar to that of DBAc-Cz and contained the featured double peak of TBD around 220–265 nm, but the absorption peaks over 300 nm were found to be wider (Figure 4b). Tauc plots were utilized to calculate the energy level from ground state to S_1_ state using the UV-vis diffuse reflection data of these samples (Figure 4c,d). DBAc-Cz’s and TBD’s band gaps were also evaluated by theoretical calculation (Appendix A) to fit the Tauc plot’s results (2.86 eV and 4.55 eV). The bandgaps of DT100, DT50, and DT20 (2.59 eV, 2.46 eV, and 2.25 eV, respectively) were lower than that of DBAc-Cz, while the bandgap of DT500 (2.98 eV) was slightly higher than that of DBAc-Cz. If there is no exciplex formation, the bandgaps of DT100, DT50, and DT20 should be higher than that of DBAc-Cz due to the presence of TBD. This indicated that exciplex was formed in DT along with the degeneracy of energy levels. Exciplex generated a smaller S_1_ state energy level than both DBAc-Cz and TBD, and therefore, DT showed distinct fluorescence peaks in powder state (Figure 2b) and exhibited different emission peaks around 505–515 nm in DCM compared to the DBAc-Cz solution (Figure 4e). These emission peaks were considered from the exciplex within DT. As for the unusual large bandgaps of DT500, it was attributed to the high TBD content which had a large intrinsic bandgap of 4.55 eV (Figure 4c).

When DT were solved in DCM, the H-bonds were broken, and the intermolecular distance between DBAc-Cz and TBD became larger. Some exciplex separated and resulted in the reappearance of the DBAc-Cz’s characteristic fluorescence peak in the DT DCM solution. For example, the intensity of the exciplex’s emission peaks of DT50 in DCM reduced with the decreasing concentration. When the concentration dropped to 0.0002 g/mL, the emission of the DT50 solution was identical to the blue-shifted DBAc-Cz solution’s emission because the intermolecular distance was too large to form exciplex (Figure 4f). The same phenomenon was also observed in the steady-state PL spectra for the DT500, DT100, and DT20 DCM solution, which also indicated the formation of exciplex in DT (Appendix A).

When DBAc-Cz and TBD were ground together, intermolecular H-bonds formed between the two compounds, and exciplex self-assembled. The exciplex can be regarded as the minimum emission unit in DT, which explained why DBAc-Cz’s emission characteristic peak did not appear in the emission spectra of DT. The H-bonds not only acted as the requirement of the formation of exciplex but restricted the vibration and rotation of DBAc-Cz and TBD, limiting the nonradiative transition of triplet excitons. DBAc-Cz itself has the potential to emit phosphorescence. PVA film embedded with DBAc-Cz can emit RTP for a visible afterglow of about 7s (Appendix A), which confirmed DBAc-Cz’s ability to generate triplet excitons. When TBD is excited, the p-π conjugate structure with three N atoms is more likely to generate triplet excitons because the N atoms with lone pair of electrons conduce to spin-coupling and ISC process. TBD’s large energy bandgap of HOMO to LUMO makes it possible to transfer energy to the exciplex through Dexter excitation transfer. Favorable triplet exciton generation and limitative nonradiative relaxation led the exciplex to beam efficient RTP with long lifetime and afterglow.

Enabled by the H-bond interaction within the exciplex, water-stimulated double emission ink was developed. The main composition of paper is cellulose abounding with hydroxy groups. It can serve as a H-bond matrix of DT. To realize the water-stimulated photoluminescent ink, a filter paper with 0.01 g/mL DT50 DCM solution (noted as DT ink) was submerged in water and then dried at room temperature. The resulting paper emitted fluorescence and RTP under 365 nm UV radiation (Figure 5b,c). The DT50-treated paper was found to be the most efficient fluorescence and RTP material among all the tested DT. After the DT-ink-treated paper was tried, the H-bonds of exciplex attached to paper were affected, resulting in the alteration of fluorescence emission and the disappearance of the RTP emission (Figure 5a–c). To demonstrate the capability of the developed technology, specific patterns were written and displayed via DT ink and water on a filter paper. The characters “USTB” were written on DT-treated paper with a pen using water as the ink. Under daylight, no character could be seen on the sample. However, under UV light, the USTB on the sample emitted different colors of fluorescence and emitted little RTP emission when the UV lamp was off (Figure 5d). The water-written area of the sample showed distinct fluorescent and phosphorescent behavior but showed no difference under daylight. Because the “USTB” characters were invisible under daylight but clearly visible under 365 nm UV light and after the removal of UV light, the encryption and decryption of the characters were successfully achieved. The developed DT ink showed great potential in photoluminescent double-encryption applications.

## 3. Materials and Methods

### 3.1. General Information

All the solvents are purchased from Aldrich and Peking Reagent and used without any further purification. 2,7-dibromo-9H-carbazole, 4-(methoxycarbonyl)phenylboronic acid, 1,5,7-Triazabicyclo [4.4.0]dec-5-ene (TBD) and Palladium(II)acetate (Pd(O_2_CCH_3_)_2_) are purchased from Leyan company. Sodium hydroxide (NaOH), sodium sulfate anhydrous (Na_2_CO_3_), sodium sulphate anhydrous (Na_2_SO_4_) and hydrochloric acid 38wt% (HCl) are purchased from Macklin.

1. H NMR spectra are obtained on Bruker Avance 400 MHz NMR spectrometers using chloroform-d (CDCl_3_) and dimethyl sulfoxide-d6 (C_2_D_6_SO) as solvent and tetramethylsilane (TMS) as the internal standard. High-resolution mass spectra (HRMS) are obtained on Micromass GCT-MS instrument operating in Matrix-Assisted Laser Desorption Ionization (MALDI) mode and time of flight (TOF) mass detector. Ultraviolet-visible (UV-vis) absorption spectra are measured by V-570 UV/VIS/NIR spectrophotometer at room temperature. UV-vis diffuse reflectance spectra are measured using Hitachi model UH4150 UV-vis-NIR spectrophotometer. Photoluminescence spectra (excitation/emission spectra, delayed spectra, decay curve and quantum yield are measured by FLS920 and FLS1000 Photoluminescence Spectrometer at room temperature. Infrared spectra are measured by Perkin Elmer LR-64912C FT-IR spectrometer.

### 3.2. Synthesis of 4,4’-(9H-Carbazole-2,7-diyl)Dibenzoic Acid (DBAc-Cz)

0.360 g (2 mmol) of 4-(methoxycarbonyl)phenylboronic acid and 0.325 g (1 mmol) of 2,7-dibromo-9H-carbazole were mixed with 50 mL ethanol (EtOH) and 50 mL of 0.2 M Na_2_CO_3_ solution in round-bottom flask. The solution was deoxidized by ultrasound in argon gas atmosphere. Then, 50 mg of Pd(O_2_CCH_3_)_2_ was added into solution as the catalyst of this Suzuki-Miyaura coupling reaction. The mixture was stirred for 24 h in 363k (90 °C) under an argon atmosphere. The product was extracted into dichloromethane (DCM) from reaction solution and was purified by column chromatography (ethyl acetate/petroleum ether) to obtain the intermediate product (DBzMe-Cz). 0.40 g DBzMe-Cz was hydrolyzed in 50 mL dioxane and 30 mL 1 M NaOH solution for 8 h in 373k (100 °C) while refluxing. HCl (aq) was added to reaction solution until the pH turned to 1–2 to supersaturate DBAc-Cz as brown floccules. The sediment was centrifuged and washed with deionized water for three times. Then, the sediment was dried in vacuum at 333 k (60 °C) to obtain DBAc-Cz.

DBAc-Cz: ^1^H NMR (400 MHz, DMSO-*d*_6_) δ 12.98 (s, 2H), 11.39 (s, 1H), 8.23 (d, *J* = 8.2 Hz, 1H), 8.16 (d, *J* = 7.9 Hz, 1H), 8.06 (d, *J* = 7.7 Hz, 4H), 7.90 (d, *J* = 9.1 Hz, 4H), 7.80 (s, 1H), 7.53 (t, *J* = 7.8 Hz, 3H), 7.41 (t, *J* = 7.7 Hz, 1H), 7.18 (dd, *J* = 10.6, 4.8 Hz, 1H) (Appendix A).

HRMS (MALDI-TOF, *m/z*): calcd for C_26_H_17_NO_4_, 407.43, found: 407.03.

### 3.3. Synthesis of 4,4’-(9H-Carbazole-2,7-diyl)Diphenol (DPOl-Cz)

Synthesis route of DPOl-Cz is similar as the route of DBAc-Cz. 0.276 g (2 mmol) of 4-hydroxyphenylboronic acid and 0.325 g (1 mmol) of 2,7-dibromo-9H-carbazole were mixed with 50 mL ethanol (EtOH) and 50 mL of 0.2 M Na_2_CO_3_ solution in round-bottom flask. The solution was deoxidized by ultrasound in argon gas atmosphere. Then, 50 mg of Pd(O_2_CCH_3_)_2_ was added into solution as the catalyst of this Suzuki-Miyaura coupling reaction. The mixture was stirred for 24 h in 363k (90 °C) under argon gas atmosphere. HCl (aq) was added to reaction solution to neutralize Na_2_CO_3_ till pH turned to 5–6. The product was extracted into dichloromethane (DCM) from reaction solution then evaporated the solvent under reduced pressure and dried the crude product. 0.40 g crude product was dissolved in 40 mL dioxane and 20 mL 1 M NaOH solution. HCl (aq) was added to the solution until the pH turned to 1–2 to supersaturate DPOl-Cz as black-brown floccules. The sediment was centrifuged and washed with deionized water and dioxane (10:1 volume ratio) for three times. The sediment was dried in vacuum at 333 k (60 °C) to obtain DPOl-Cz.

DPOl-Cz: ^1^H NMR (500 MHz, DMSO-*d*_6_) δ 11.23 (s, 1H), 9.52 (s, 2H), 8.11 (s, 1H), 8.10 (s, 1H), 7.61 (d, *J* = 1.6 Hz, 2H), 7.58 (d, *J* = 2.1 Hz, 2H), 7.57 (d, *J* = 2.1 Hz, 2H), 7.39 (d, *J* = 1.6 Hz, 1H), 7.37 (d, *J* = 1.6 Hz, 1H), 6.90 (d, *J* = 2.0 Hz, 2H), 6.88 (d, *J* = 2.2 Hz, 2H) (Appendix A).

HRMS (MALDI-TOF, *m/z*): calcd for C_24_H_17_NO_2_, 351.41; found: 351.16.

### 3.4. Preparation of DT

TBD was purified by recrystallization in THF. 0.1395 g TBD and 0.0008 g/0.0040 g/0.0080 g/0.0202 g DBAc-Cz was placed into a agate mortal. Ground the two kinds of powder with force thoroughly for 2 min to produce canary yellow powder of DT500/DT100/DT50/DT20.

## 4. Conclusions

In summary, a new carbazole derivative DBAc-Cz was synthesized and was combined with TBD to produce H-bond photoluminescent materials (DT) using a simple grinding method. DBAc-Cz and TBD were linked by H-bonds to form exciplex within DT so the fluorescence and phosphorescence behavior changed significantly. The prepared DT exhibited efficient RTP emission with a 0.85 s lifetime and a quantum yield of 11.71%. The minimal difference between fluorescence and RTP emission wavelength of DT is only 14 nm (DT50). This work provides insights into the development of stimulus-responsive exciplex RTP materials. Water-stimulated DT ink with double emission was developed, showing potential application in the writing and printing encryption field.

## Figures and Tables

**Figure 1 molecules-27-06482-f001:**
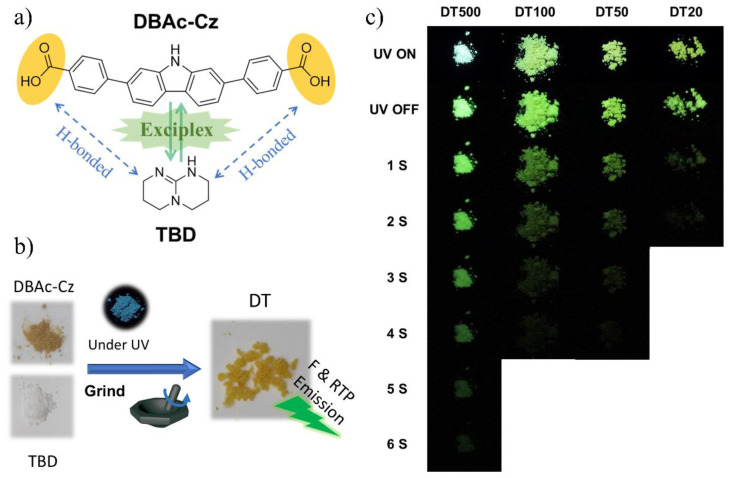
(**a**) Chemical structure of DBAc-Cz and TBD and the formation of exciplex between the two molecules by H-bonds. (**b**) Photographs of DBAc-Cz, TBD, DT and excited DBAc-Cz with UV irradiation (F-fluorescence). (**c**) Photographs of excited DTs after removal of UV irradiation. Note: all the UV irradiation is at 365 nm.

**Figure 2 molecules-27-06482-f002:**
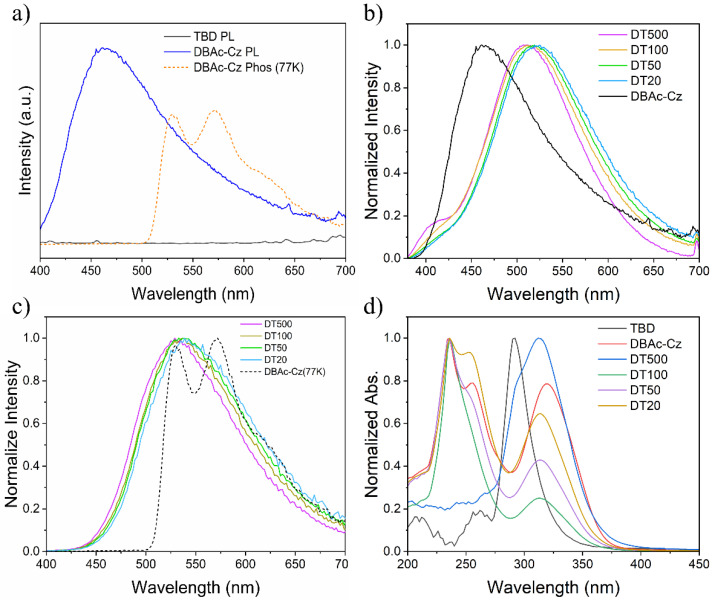
Photoluminescence characterization and UV-vis absorption spectra. (**a**) Steady-state PL spectra (solid line) of TBD and DBAc-Cz, phosphorescence spectra (delay 50 ms) at 77 K (dash line) of DBAc-Cz. (**b**) Normalized steady-state PL spectra of DT500, DT100, DT50, DT20, and DBAc-Cz. A continuous xenon lamp was used as the excitation light source. (**c**) Normalized phosphorescence spectra at room temperature (RTP) for DT500, DT100, DT50, DT20 and phosphorescence spectra at 77 K for DBAc-Cz. A pulse xenon lamp was used as the excitation light source. (**d**) Normalized UV-vis absorption spectra of TBD, DBAc-Cz, DT500, DT100, DT50 and DT20 in dichloromethane (DCM) solution.

**Figure 3 molecules-27-06482-f003:**
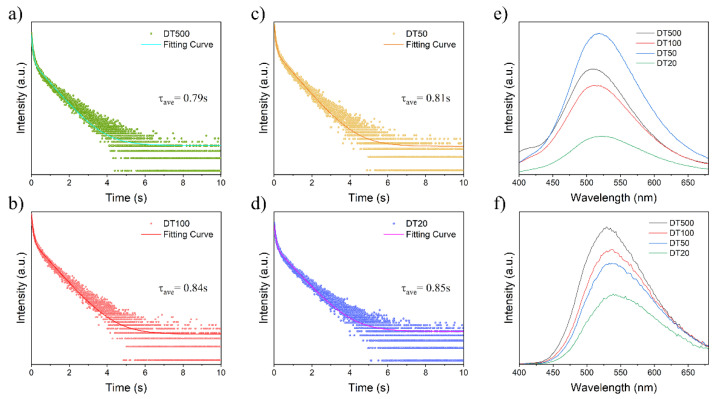
Room temperature phosphorescence decay curve and lifetime (**a**) DT500, (**b**) DT100, (**c**) DT50 and (**d**) DT20. (**e**) Steady-state PL spectra of DTs measured using the same device parameters and ambient conditions. (**f**) Phosphorescence spectra of DTs measured using the same device parameters and ambient condition.

**Figure 4 molecules-27-06482-f004:**
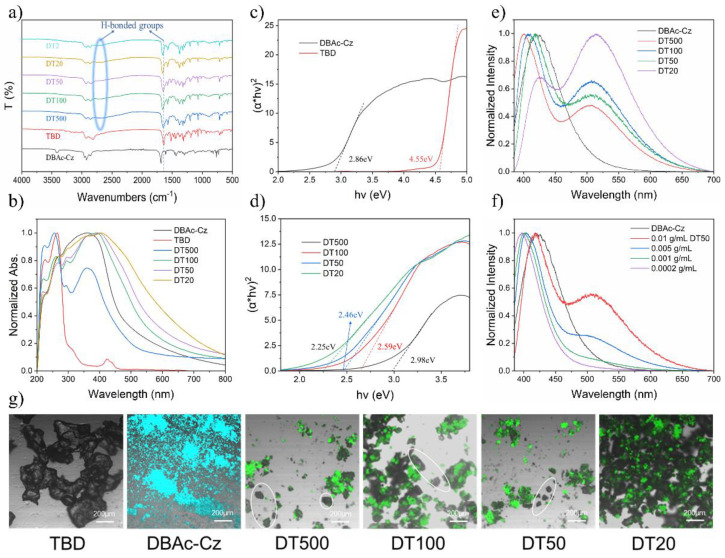
(**a**) FT-IR spectra and (**b**) UV-vis diffuse reflection spectra of DBAc-Cz, TBD, DT500, DT100, DT50, DT20 (DT2: DBAc-Cz/TBD = 1/2 mole ratio). Tauc plots and the calculated band gaps of (**c**) DBAc-Cz, TBD and (**d**) DT. (**e**) Steady-PL spectra of DT solution in DCM (0.01 g/mL). (**f**) Steady-PL spectra of different density (0.01 g/mL, 0.005 g/mL, 0.001 g/mL, 0.0002 g/mL) of DT50 solution in DCM. (**g**) Photographs of TBD, DBAc-Cz and DT (excited by 405 nm UV), non-photoluminescent particles were marked by white ellipse.

**Figure 5 molecules-27-06482-f005:**
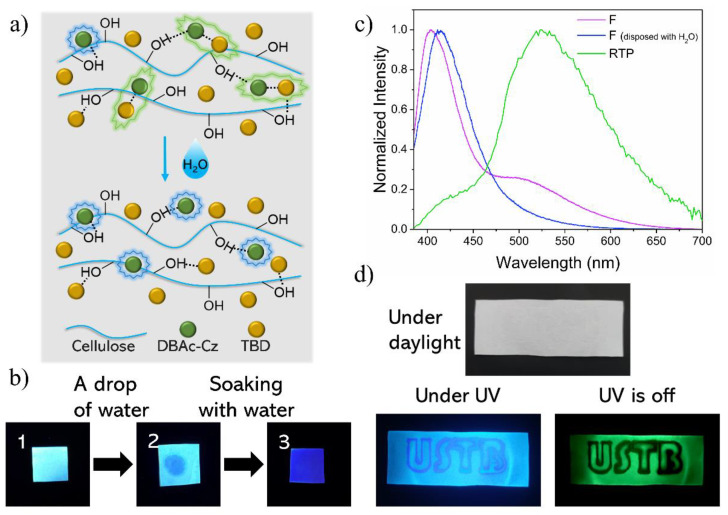
(**a**) Schematic illustration of DT embedded in paper (cellulose) and after water-processed. (**b**) Graphs of samples under 365 nm UV lamp. DT ink infiltrated paper (1), DT ink infiltrated paper with a drop of water on it and then dried (2), DT ink infiltrated paper soaked with water and then dried (3). (**c**) Steady-PL spectra of DT ink infiltrated paper, DT ink & water infiltrated paper and phosphorescence spectra of DT ink disposed filter paper (F: fluorescence, RTP: room temperature phosphorescence). (**d**) DT ink infiltrated paper was written with water to achieve word encryption. Graphs of the sample under daylight, under and after the removal of 365 nm UV light.

**Table 1 molecules-27-06482-t001:** Photophysical properties of the components and DT.

Sample	Em_f_(max) (nm)	Em_p_(max) (nm)	τ_RTP_(s)	Difference of Em_p_(max) and Em_f_(max) (nm)
DBAc-Cz	461	528/572 (77 k)	/	67/111
DBAc-Cz (PVA) *	414	519	0.52	105
TBD	/	/	/	/
DT500	507	529	0.79	22
DT100	511	537	0.81	26
DT50	519	533	0.84	14
DT20	525	539	0.85	16

* DBAc-Cz (PVA) means PVA film doped with DBAc-Cz, excitation wavelength: 365 nm.

## Data Availability

Not applicable.

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
