# Peer review of "H-Bonding Room Temperature Phosphorescence Materials via Facile Preparation for Water-Stimulated Photoluminescent Ink"

_molecules, 2022, doi:10.3390/molecules27196482_

Round 1

Reviewer 1 Report

In this manuscript, Yang and co-workers report an interesting study of room temperature phosphorescent materials. An ingenious design strategy for noncovalent room temperature phosphorescent materials is developed by mixing a carbazole derivative 4,4'-(9H-carbazole-2,7-diyl)dibenzoic acid and 1,5,7-Triazabicyclo[4.4.0]dec-5-ene. The formation of hydrogen bonds between the building blocks is an excellent driving force to generate and enhance phosphorescence. Furthermore, such dual-emissive phosphorescent materials realize 0.85 s lifetime and 11.71% quantum yield, and are used as water-stimulated ink. Therefore, I would recommend this manuscript to be published on Molecules after minor revisions:

1, In Figure 4a, the FT-IR spectra of DT500 should be given here, and page 6, line 170-175, the author should give the corresponding reference to support this statement.

2, In Figure 4g, why is the non-photoluminescent DT50 particles less than DT100?

3, Page 8, line 236-237, why is DT50 chosen as water-stimulated photoluminescent ink rather than other DT materials?

4, The authors may consider citing the latest progress on smart noncovalent photoluminescent materials, such as Nat. Commun. 2022, 13, 3216; Matter 2022, 5, DOI: https://doi.org/10.1016.j.matt.2022.07.022.

Author Response

1, In Figure 4a, the FT-IR spectra of DT500 should be given here, and page 6, line 170-175, the author should give the corresponding reference to support this statement.

Author reply: Thanks for this valuable suggestion. The FT-IR spectra of DT500 is given in Figure 4a. Corresponding references are added to support this statement.

2, In Figure 4g, why is the non-photoluminescent DT50 particles less than DT100?

Author reply: Thanks for this valuable suggestion. One possible explanation is because the content of DBAc-Cz in DT50 is higher than DT100, the more photoluminescent particles are formed in DT50, so the non-photoluminescent particles of DT50 is less than DT100. The other explanation is that the DT is not homogeneous by grinding method. Considering the inconsistent size and shape of particles, the quantitative relationship is difficult to determine.

3, Page 8, line 236-237, why is DT50 chosen as water-stimulated photoluminescent ink rather than other DT materials?

Author reply: Thanks for this valuable suggestion. DT materials have all been tested as water-stimulated photoluminescent ink. Among them DT50 treated paper was found to have the most efficient visible fluorescence and RTP. It is described in the manuscript at line 241-243. So, DT 50 is chosen as water-stimulated photoluminescent ink.

4, The authors may consider citing the latest progress on smart noncovalent photoluminescent materials, such as Nat. Commun. 2022, 13, 3216; Matter 2022, 5, DOI: https://doi.org/10.1016.j.matt.2022.07.022.

Author reply: Thanks for this valuable suggestion. The related references and necessary statements have been updated in the manuscript to make the paper more comprehensive.

For more details, please check the attachment of our response letter.

Reviewer 2 Report

The manuscript is dedicated to pure organic room-temperature phosphorescence (RTP) material based on 4,4'-(9H-carbazole-2,7-diyl)dibenzoic acid (DBAc-Cz) and 1,5,7-Triazabicyclo[4.4.0]dec-5-ene (TBD). Exciplex forms between the molecules due to the H-bonding. The room-temperature phosphorescence comes from the exciplex. The paper is well written, and all emission mechanisms are proven. However, a deeper explanation could be given about time-resolved measurements and data processing.

The approximation line does not correctly represent data points because it does not pass through the middle part of the experimental points but is more shifted towards the lower part. Probably more a careful approximation of the data points could be done.

The paper could be accepted in the journal Molecules after minor revision.

Author Response

Author reply: Thanks for this valuable suggestion. However, in this article, approximation line is not adopted in any Figures in the manuscript or supporting information. Fitting curves are applied in Figure 3 to calculate the lifetime of RTP emission. This is a calculation method rather than the approximation process of data. In photoluminescent research, decay curve with the fitting curve and calculation is a common approach to measure the lifetime for fluorescence or phosphorescence emission. For the consistency of the manuscript, we didn’t give more explanation about time-resolved measurements and data processing.
